# Content Validation of a Chrononutrition Questionnaire for the General and Shift Work Populations: A Delphi Study

**DOI:** 10.3390/nu13114087

**Published:** 2021-11-15

**Authors:** Yan Yin Phoi, Maxine P. Bonham, Michelle Rogers, Jillian Dorrian, Alison M. Coates

**Affiliations:** 1UniSA Allied Health and Human Performance, University of South Australia, Adelaide, SA 5001, Australia; yan_yin.phoi@mymail.unisa.edu.au; 2Alliance for Research in Exercise, Nutrition and Activity (ARENA) Research Centre, University of South Australia, Adelaide, SA 5001, Australia; michelle.rogers@unisa.edu.au; 3Department of Nutrition, Dietetics and Food, Monash University, Melbourne, VIC 3168, Australia; maxine.bonham@monash.edu; 4UniSA Justice and Society, University of South Australia, Adelaide, SA 5072, Australia; jill.dorrian@unisa.edu.au; 5Behaviour-Brain-Body Research Centre, University of South Australia, Adelaide, SA 5072, Australia

**Keywords:** chronotype, circadian rhythm, meal regularity, meal timing, temporal meal patterns, questionnaire development, nutrition assessment

## Abstract

Unusual meal timing has been associated with a higher prevalence of chronic disease. Those at greater risk include shift workers and evening chronotypes. This study aimed to validate the content of a Chrononutrition Questionnaire for shift and non-shift workers to identify temporal patterns of eating in relation to chronotype. Content validity was determined using a Delphi study of three rounds. Experts rated the relevance of, and provided feedback on, 46 items across seven outcomes: meal regularity, times of first eating occasion, last eating occasion, largest meal, main meals/snacks, wake, and sleep, which were edited in response. Items with greater than 70% consensus of relevance were accepted. Rounds one, two, and three had 28, 26, and 24 experts, respectively. Across three rounds, no outcomes were irrelevant, but seven were merged into three for ease of usage, and two sections were added for experts to rate and comment on. In the final round, all but one of 29 items achieved greater than 70% consensus of relevance with no further changes. The Chrononutrition Questionnaire was deemed relevant to experts in circadian biology and chrononutrition, and could represent a convenient tool to assess temporal patterns of eating in relation to chronotype in future studies.

## 1. Introduction

Circadian rhythms are daily rhythms in our body that repeat every 24 h; they determine our behaviour, such as when we wake and sleep, as well as the times at which metabolic processes occur within our body [1,2]. These rhythms run even in the absence of external cues and are regulated internally by a “central clock” in the Suprachiasmatic Nucleus (SCN) in synchrony with other “peripheral clocks” in the body, in relation to the 24 h cycle of light and dark [2,3].

Eating and sleeping at irregular hours in relation to the light–dark cycle results in misalignment between central and peripheral clocks, or circadian misalignment [4], which has shown to negatively impact multiple physiological markers including raised blood pressure and elevated inflammation [5]. Circadian misalignment affects up to 1.4 million shift workers in Australia [6], who work outside of usual weekday work hours of 9 a.m. to 5 p.m. [7], and to a lesser extent, the 20% of the adult population who identify as an evening chronotype [8]. Evening chronotypes tend to wake and sleep late [9], with increasing numbers in society as individuals stay engaged with technology late into the night [10]. Importantly, from a dietary perspective, evening chronotypes and shift workers tend to eat later in the day [11,12,13]. Eating later rather than earlier resulted in higher blood glucose levels [14,15] and perturbed lipid metabolism [16]. Over a prolonged period of years, these metabolic disturbances could be a factor in the increased risk of obesity, diabetes, and cardiovascular disease often observed amongst these population groups [17,18,19,20].

Chrononutrition is the understanding that timing of food consumption interacts with internal circadian rhythms to impact on health outcomes [21] and encompasses not only the frequency and regularity of eating behaviour [22], but also the duration and timing of the eating window (the period between the first and last time of calorie consumption in a day) [23]. Time-related aspects of eating—hereafter collectively referred to as temporal patterns of eating—are the subject of a recent position statement by the American Heart Association, where a link with cardiometabolic health has been outlined [24]. Yet, to date, multiple tools are needed to evaluate these concepts, with no single tool available to accurately capture these temporal patterns of eating [22]. As such, development of a Chrononutrition Questionnaire that enables identification of temporal patterns of eating in relation to chronotype in the general and shift work populations is warranted. Such a tool will enable temporal patterns of eating to be mapped against each individual’s circadian rhythm and further investigation into disease risk.

### Development of a Chrononutrition Questionnaire

The first phase of tool development is item development, consisting of: (i) domain identification and item generation and (ii) establishing content validity [25]. Domain identification and item selection was the subject of a recent scoping review, which evaluated the impact of chronotype on temporal patterns of eating whilst simultaneously identifying and assessing the strengths and limitations of existing tools used for data collection [26]. Our scoping review found inconsistencies between studies with relation to using validated dietary tools and an inability to capture all aspects of temporal meal patterns and its relation to chronotype [26]. As such, a single standardised and validated tool to capture these outcomes of interest, ensuring consistency in data interpretation in future studies, is required.

In addition, our review informed item generation and identified eight categories (domains) of temporal eating patterns consistent with terminology associated with chrononutrition, which in turn generated seven preliminary outcomes for inclusion in the Chrononutrition Questionnaire (Figure 1). An additional component of the scoping review, important to include in any new questionnaire, is the identification of chronotype. Historically, the Munich Chronotype Questionnaire (MCTQ) is the preferred choice, as it identifies differences in individual circadian rhythm cycle, specifically through mid-sleep time on work-free days, corrected for sleep debt over the work week (MSF_SC_) [27]. Furthermore, a version of the MCTQ that accounts for their varying shift schedules and sleep habits of shift workers has also been validated (MCTQ^Shift^), allowing them to be chronotyped more accurately [28]. As such, incorporation of components of the MCTQ in order to determine sleep time, wake time, and sleep duration is an important inclusion. As the Chrononutrition Questionnaire should be applicable to the general and shift work populations, the seven identified outcomes will encompass different scenarios (e.g., workdays, school days, work/school-free days).

Thus, this research forms the second part of item development—the Delphi survey method. This methodology establishes content validity by putting experts who are representatives of the field of study through a series of survey “rounds” that obtain their ratings and comments about an issue to achieve consensus [29]. Its unique characteristics include: (i) controlled feedback, of individual and group responses to each expert after every round; (ii) statistical group response, where responses from each round are statistically analysed and summarised by the researcher; (iii) iteration, where each expert is allowed to adjust his/her answers after considering the group response from the previous round; and (iv) anonymity, of the identity behind each expert’s response among the expert panel [29,30,31]. Beyond consensus, it generates discussion and exploration of ideas amongst experts in the field [32].

## 2. Materials and Methods

The reporting of this paper follows the recommendations for the Conducting and Reporting of Delphi Studies (CREDES) [33]. In addition, we follow the standard set of quality indicators in reporting Delphi methodology proposed by Diamond et al. [34], and our criteria are summarised in Table 1.

### 2.1. Recruitment

Participants were recruited through non-probability purposive sampling to identify experts in the area of circadian biology and chrononutrition who may provide meaningful feedback, followed by snowball sampling, which allowed initially identified experts to suggest and refer the study to suitable peers. There are no guidelines on the ideal number of experts to serve on a panel. Sample size is often based on identifying experts who have knowledge of the subject area and able to contribute meaningfully to the Delphi rounds, and based on rigorous inclusion and exclusion criteria [35]. As circadian biology and chrononutrition is a relatively specialised field, a homogenous sample was expected. Hence, a target of 10–15 experts was chosen, which would allow for results to be representative of a larger population [36]. For this Delphi study, participant eligibility was based on the inclusion and exclusion criteria listed in Table 1.

### 2.2. Data Collection and Management

Study data provided by participants at screening, consent, and Delphi rounds were collected and managed using REDCap (version 11.1.14, Vanderbilt University, Nashville, TN, USA, Available online: https://research.unisa.edu.au/redcap/ (accessed on 1 December 2020), a secure, web-based software platform designed to support data capture for research studies, hosted at the University of South Australia [37,38].

All identified experts were contacted by email, and foll owing research guidelines for the Delphi survey method, were informed of the study expectations of them, the time commitment required, and the overarching purpose of their contribution in order to foster relationships and encourage optimal responses across Delphi rounds [32]. The email also contained a hyperlink that brought them through screening for eligibility and provision of consent. Experts who were eligible and provided consent were immediately directed to the 1st round of the Delphi survey. They were given two weeks to complete each round, with reminder emails sent out on the first and last working day of the 2nd week to those who did not respond to the first email invitation of each round. Experts who declined to participate upon initial email contact or who did not participate in the 1st round of the Delphi survey were excluded and not invited to subsequent rounds. Experts who participated in the 1st round but not the 2nd round were deemed as loss to follow-up and were not contacted for the 3rd round. The Delphi rounds were completed between January and March 2021.

In the 1st round of the Delphi survey, demographics of experts were collected, including gender, age, highest level of education, role (academic/clinical), years of experience in the field of circadian biology or chrononutrition, and country of work. Experts were provided information on the background of the Chrononutrition Questionnaire, followed by the questionnaire itself, including instructions and definitions of terms that would be provided to end users. The Chrononutrition Questionnaire consisted of 46 questions that covered the seven main outcomes (Figure 1) for each “day type” (e.g., work, school, work-free days, and different shift types for shift workers) experienced by potential respondents. The question formats included a mix of Yes/No answers and free text boxes for participants to fill in clock times. All materials provided to the expert panel were developed by YP following the scoping review and were reviewed, discussed, and edited in consultation with co-authors M.R., A.C., J.D., and M.B.

As content validity requires relevance of content [39], experts were asked to rate the relevance of each question on a Likert scale, a common method used in Delphi studies [40]. A 4-point Likert scale was used, as neutral answers are not conducive to meet the goal of clarifying opinions [41]. The scale had the following range: 1 = irrelevant, 2 = relevant with major changes, 3 = relevant with minor changes, and 4 = relevant with no changes [42]. A free text box was provided with each question and at the end of the questionnaire to allow for comments or rationale for their choice. These free text responses could be about, but were not limited to, (i) item content: the question’s representation of the content; (ii) item style: clarity of the question’s wording to accurately measure the construct; and (iii) comprehensiveness: questionnaire completeness in measuring all aspects of the construct [43]. Experts were also encouraged to suggest any additional questions that they perceived to be relevant for inclusion in the Chrononutrition Questionnaire, which were subject to review by the expert panel in subsequent rounds.

For this study, a consensus level of ≥70% was chosen to guide changes to the questionnaire, in line with other research on questionnaire development in the area of health [42,44]. Based on ≥70% consensus, questions rated “1” were removed and “4” accepted. Those questions with a ≥70% consensus of “4” were still included in subsequent rounds for evaluation if they had been edited based on comments from the expert panel in the previous round. Questions that did not gain a ≥70% consensus rating of “4”, but whose sum of ratings of “2” and “3” was greater than ratings of “1”, were amended based on comments and evaluated in the next round.

At the start of the subsequent round, experts were contacted via email and provided with (i) analytical statistics of ratings for each question, (ii) summary of comments grouped into common themes, (iii) ad verbatim comments, and (iv) a hyperlink and unique password to access that round. Once in the next round, they were able to see the revised Chrononutrition Questionnaire, with additions or changes made to it based on expert feedback from the previous round indicated; comments that were not incorporated into the questionnaire had justifications provided.

While achieving consensus is often the reason for terminating the Delphi process [34], it is not a requirement for stopping. Systematic reviews suggest two or three rounds occur in 90% of Delphi studies [34] and this is the number of rounds that is commonly recommended [40]. Response exhaustion tends to occur after two to four rounds [35]. Therefore, the stopping criterion for this Delphi study was set at three rounds. For items where consensus of ≥70% was not reached after three rounds, results will be displayed and outcomes discussed, bearing in mind that each Delphi round had allowed for differing opinions to be documented and canvassed.

### 2.3. Data Analysis

Overall ratings for each question in the Delphi survey were analysed for percentage agreements for each level of rating. Comments provided by experts were compiled in an Excel spreadsheet and consolidated into discussion points by YP; comments between experts that overlapped were presented as one discussion point. At the end of each round, Y.P., A.C., J.D., and M.B. went through each of the discussion points to decide on resultant changes to be made to the questionnaire for the next round, as well as justifications for changes not made exactly according to the experts’ comments.

## 3. Results

Figure 2 depicts the flow chart of the number of experts involved in the Delphi survey across the three Delphi rounds. A total of 64 experts in the field of circadian biology or chrononutrition were invited to participate in the study; 59 were initially identified through non-probability purposive sampling, while five people were referred on by the experts in the initial list as suitable peers through subsequent snowball sampling. Of the 64 experts, 28 participated in Round 1, a response rate of 44%. Participants were mostly female (82%), 30–39 years old (36%), academics (100%), with a Doctorate degree (93%), with 6–10 years of experience in the field of circadian biology or chrononutrition (29%), and working in Australia (39%). Their demographic data are presented in Table 2.

### 3.1. Delphi Round 1

In Round 1, 26 experts provided full responses, and two partial responses. Figure 3 depicts the process of reviewing, amending, merging, or deleting questions based on their ratings of relevance. This round had seven main sections (each pertaining to an outcome as summarised in Figure 1) made up of 46 questions in total; seven questions achieved ≥70% consensus for the rating “relevant with no changes” (Figure 4). Table 3 provides details of the comments made and response from the research team. No outcomes reached consensus for being “irrelevant”. However, experts suggested that separate questions about “Wake time” and “Sleep time” be combined so participants may think of them in relation to each other, which simplified completing the questionnaire. The experts also recommended that “Time of first eating occasion” and “Time of last eating occasion” be removed as they could be captured more simply by “Time of main meals and snacks”. Capturing meal regularity generated much discussion. Particularly, there was a call to define “regular” and to determine regularity using a Likert scale instead of a Yes/No answer. In terms of identifying one’s largest meal, experts suggested considering calorie content instead of portion size, and capturing two or more equally large meals.

Questions specific to shift workers also received some attention. This included clarification on why sleep/wake patterns of shift workers were asked only of limited shift situations. Experts also urged that in framing questions about timing of food intake, consider that they could be affected by day-to-day variations as well as the previous day’s shift type. Furthermore, in relation to capturing temporal patterns of eating during each shift type, a time window of interest was requested to be defined (e.g., 0000 h to 2359 h or flanked by wake and sleep times before and after the shift, respectively).

Lastly, additional feedback sought to improve questionnaire format and usability by adding clear instructions to participants at the beginning, fine-tuning questionnaire wording and definition of terms, and adjusting the flow of questions. Data on participants’ work and school schedule were also requested to be collected. Additional outcomes not currently included within this questionnaire were proposed, such as sleep factors, duration of eating occasions, diet composition, and temporal variations in food-related sensations such as hunger, appetite, and satiety. At the end of the questionnaire, seven experts commented on the value and importance of such a questionnaire for the field of chrononutrition.

### 3.2. Delphi Round 2

In total, 28 experts were invited to Round 2 and 26 experts responded (24 fully, two partially), an attrition rate of 7%.

In response to the previous round, two sections were added—“Instructions to participants” and “Demographics”—to collect information about participants’ work and school schedule (adding four questions). Following recommendations from Round 1, some of the seven sections in Round 1 were merged, leaving four sections. In total, there were six sections consisting of 30 questions for experts to rate and comment on; 24 questions achieved ≥70% consensus for the rating “4” (Figure 4). Figure 3 depicts the fate of questions based on their ratings of relevance. The comments made in Round 2, along with changes and justifications, are summarised in Table 4. As before, no outcomes reached consensus for being irrelevant. “Time of largest meal” was merged with “Time of main meals and snacks”, as this was deemed easier for the participants’ thought process and ease of filling in the questionnaire. Compared to the previous round, the concept of regularity reached a higher level of consensus of relevance (65–72%), and remaining comments came from individual experts. In terms of the research team’s decision to capture temporal patterns of eating during shifts within set time limits specific to the shift (e.g., between 12 a.m. and 12 a.m. for morning shifts), there were calls for the time window to instead be limited by wake and sleep times before and after the shift.

Other comments from the expert panel continued to fine-tune the definitions of terms and choice of wording within the questionnaire to improve ease of understanding by end users. The questionnaire format was also suggested to be tweaked such that more expansive pathway questions could guide individuals to questions relevant to their unique work and/or school (for those who are studying) schedule. Lastly, one expert asked for data on diet composition to be captured alongside outcomes in this questionnaire.

### 3.3. Delphi Round 3

There were 26 experts invited to participate in Round 3 and a total of 24 experts responded (21 fully; three partially), providing an attrition rate of 8%.

In this round, experts rated and commented on 29 questions in total. They consisted of “Instructions to participants”; “Demographics”, which was expanded to six questions; and the questionnaire itself, where two sections from Round 2 were merged. This left five sections consisting of 29 questions for experts to rate and comment on; 28 questions achieved ≥70% consensus for the rating “4” (Figure 4). Figure 3 depicts the fate of questions based on their ratings of relevance.

In this third and final round, feedback predominantly sought to further fine-tune the questionnaire. The comments made in Round 3 and resultant changes are summarised in Table 5.

## 4. Discussion

Chrononutrition is receiving increasing attention as an important factor to consider when developing strategies to improve cardiometabolic health in shift workers. Previous research [26] and expert consensus from the current study indicate that chrononutrition measurement needs to include not only times at which food is consumed, but also information about amount consumed (e.g., main meals versus snacks), regularity of meals, and duration and timing of the eating window. Current methods that would capture this complexity, e.g., time-stamped food diaries, are burdensome. Therefore, our Delphi expert panel came together to agree on the most straightforward method for capturing key information to assess chrononutrition. A further layer of complexity in this study was that the instrument needed to be suitable for the general population, but also for shift workers. Our panel worked from 46 items in Round 1 to a streamlined set of 29 items collecting demographic details and assessing chronotype, wake and sleep times, meal regularity, timing of main meals and snacks, and timing of largest meal, reaching the required consensus levels for all of these items. The panel also developed instructions and key demographic variables. Key discussion points included the challenges of documenting eating patterns for shift workers relative to different shift types, capturing “regularity” of food intake, methods for taking into account calorie distribution across the day, and duration of eating occasions.

For each outcome of interest, end users were asked about temporal patterns of eating by day type (e.g., work/school day versus work-free/school-free day) to discern variations in eating patterns that may be influenced by differences in schedule between those types of days. Whilst this questionnaire caters to both shift and non-shift workers, discussion centred around the identification of eating habits during different shifts, which can be challenging to capture. Of particular debate, however, was the time window within which temporal patterns of eating before, during, and after each shift type should be captured. The final version of the questionnaire asked about the food intake window according to wake time before and sleep time after the respective shift. This was argued to be a more complete representation of mealtimes pertaining to the shift in question, especially for afternoon and night shifts, where 12 p.m. to 12 p.m. cut-offs may inaccurately omit eating episodes within a waking day. This concern was warranted as it has been shown that night shift workers eat around the clock [13], and rotating shift workers may have their first meal before 12 p.m. on the day they start afternoon and night shifts [45]. In addition, shorter durations between time of dinner relative to bedtime have been linked with greater levels of adiposity [46] and odds of being overweight or obese [47,48] potentially due to changes in melatonin onset affecting the inhibition of glucose-stimulated insulin secretion [15] and the nadir in resting energy expenditure during the biological night [49]. As such, the ability to collect data on the time of last meal relative to sleep time will allow future studies to determine whether the increased risk of overweight and obesity is associated only with time of last meal being close to bedtime at night, or also in the day (e.g., breakfast at 8 a.m. after a night shift, followed by bedtime at 9 a.m.).

The concept that caused the greatest debate was regularity of food intake. Although all questions reached the consensus threshold by the final Delphi round, questions that captured regularity had the lowest agreement among experts and required no further changes in Round 3. The inconsistency in agreement amongst experts is in line with disparity in the literature regarding the definition of “regular” when referring to meals [50]. Previous observational studies have determined meal regularity based on a single question in a questionnaire, asking about the frequency of consuming regular meals (i.e., Likert scale ranging from “Always” to “Never”) without providing a definition for “regular” [51,52] or based on consistency in energy intake per meal between days [53]. Other randomised controlled trials have compared meal regularity with irregularity by altering the frequency of, and duration between, eating occasions from day to day to create “chaos” in mealtimes [54,55]. Perhaps capturing the concept of regularity is challenging because while it is proposed to be linked to disruption of eating patterns, and hence the internal circadian clock, the extent of irregularity required to negatively impact on health has not yet been comprehensively explored [50].

To address these challenges, we have taken on suggestions from the expert panel to determine regularity using a Likert scale (ranging from “never” to “always” regular), with “regular” food intake defined as deviations of not more than ± 30 min from day to day. This method allows us to identify the overall “chaos” or irregularity of meals, which by disrupting the circadian rhythmicity of physiological processes, is what adversely affects cardiometabolic health [56]. This approach avoids the need to discern between regularity of each meal type or the extent of irregularity (e.g., deviations of 30, 60, or 90 min between days). Lastly, we intentionally chose to segregate regularity by day type (e.g., work/school days versus work-free/school-free days, or morning shift versus afternoon shift) to identify whether there are certain days that have a greater extent of variability.

Previous studies have suggested that capturing calorie distribution across the day is relevant, since a greater percentage of calories consumed at earlier rather than later times of the day has been found to result in greater weight loss and improved glycaemic and lipid profiles in overweight adults [57], and improved glycaemic profiles in both people with type 2 diabetes [58] and healthy adults [59]. This is supported further by studies of early time restricted eating, where limiting food intake to earlier parts of the day resulted in weight loss due to a reduction in energy intake [60], while in the absence of weight loss, saw improvements in insulin sensitivity, blood pressure [61], glucose levels, and lipid metabolism [62]. Therefore, in this study, while the “Time of main meals and snacks” was designed to capture the temporal spread of food intake across the day, with participant-identified main meals indicating the times at which most of the caloric load is consumed, the “Time of largest meal” was designed to indicate the time of the meal where caloric load is the greatest in the day.

The duration of eating occasions arose as an important aspect to capture when identifying eating occasions. This aspect is considered relevant to health outcomes, since a systematic review found slower eating rates to be associated with lower energy intake [63], while individuals who ate quickly were more likely to be overweight and obese [64,65,66], and have hypertriglyceridemia [67] and metabolic syndrome [68]. These associations are not surprising, as lack of food stimulation in the oral cavity decreases satiety while increasing the desire to eat, leading to greater food consumption [69]. Yet, the duration of eating occasions may not be solely indicative of the extent of oral stimulation and thus food intake, as it can be additionally influenced by other factors, including eating utensil used [70], texture of food [71], and social setting and company [72,73].

This Delphi study has a number of strengths. Firstly, experts in the panel were well-represented geographically and had great mastery of the subject matter, with almost half of them having >10 years of experience in the field of circadian biology or chrononutrition. Each Delphi round also had a low attrition rate, with response rates being greater than the recommendation of 70% [74]. This is coupled with a high level of participation from the expert panel, with comments in Rounds 2 and 3 referring to other experts’ opinions from the previous round. Secondly, in the final round, only one question did not achieve the consensus threshold. This question was not a main outcome related to capturing chronotype or temporal patterns of eating, but fell under “Demographics” and was concerned with inclusivity of individuals who neither have work nor school (e.g., retirees, unemployed, disabled), which we have since provided an option for. Lastly, questions that had achieved consensus in previous rounds were edited further based on comments provided and still subject to evaluation in the subsequent round. This helped to improve the relevance and clarity of questions within the questionnaire, as they reached an even higher level of consensus with each round, increasing the content validity of the questionnaire. This questionnaire has a few limitations. Firstly, shift workers are asked about sleep and wake times during very specific shift scenarios based on the MCTQ^Shift^ [28]; hence, it can only determine the temporal patterns of eating but not chronotype or sleep pattern of shift workers with certain patterns of rotating shift work or who perform split shifts. Secondly, we chose to capture the spread of energy distribution across the day based on the time of largest meal amidst times of main meals and snacks, instead of by calorie intake at set time intervals (e.g., 0900–1200 h, 1200–1500 h) across the day as in previous studies [75,76]. This approach chosen will be investigated in a subsequent study by comparison against food diaries completed within the same time frame.

## 5. Conclusions

Through the Delphi survey method, experts in circadian biology and chrononutrition provided insights based on their experience to evaluate and improve the relevance of a Chrononutrition Questionnaire. Experts agreed with ≥70% consensus that the questions, with the ability to capture mealtimes, frequency, regularity, skipping, and duration of eating window, in relation to chronotype, are relevant to the purpose of the tool. Their high level of participation highlights the importance and value of this tool, and that it can be used in future studies with different populations, such as individuals with medical conditions and shift workers. This tool has the potential to serve as a screening tool for organisations or healthcare workers to identify suboptimal temporal patterns of eating amongst employees or patients and escalate nutrition care. More importantly, analysis of questionnaire outcomes against health outcomes in future studies will inform the development of guidelines to optimise temporal patterns of eating and improve health outcomes in different populations within our society (including those working during the day as well as shift workers). The results and output from this Delphi survey can now confidently inform the next stage of questionnaire development by undergoing testing of test re-test reliability and construct validity.

## Figures and Tables

**Figure 1 nutrients-13-04087-f001:**
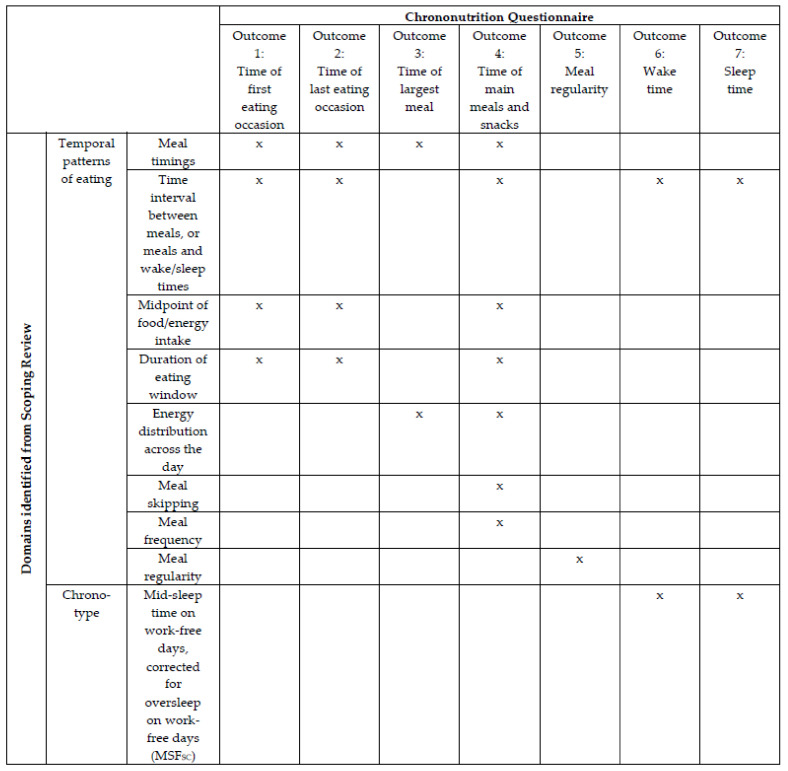
Domains identified by a scoping review (rows) and the seven preliminary outcomes that address them in the Chrononutrition Questionnaire (columns). “Largest meal” is defined based on portion size while “main meals” refer to breakfast, lunch, dinner, and/or supper. “x” represents the outcome that alone, or with other outcome(s) along the same row captures the domain on that row.

**Figure 2 nutrients-13-04087-f002:**
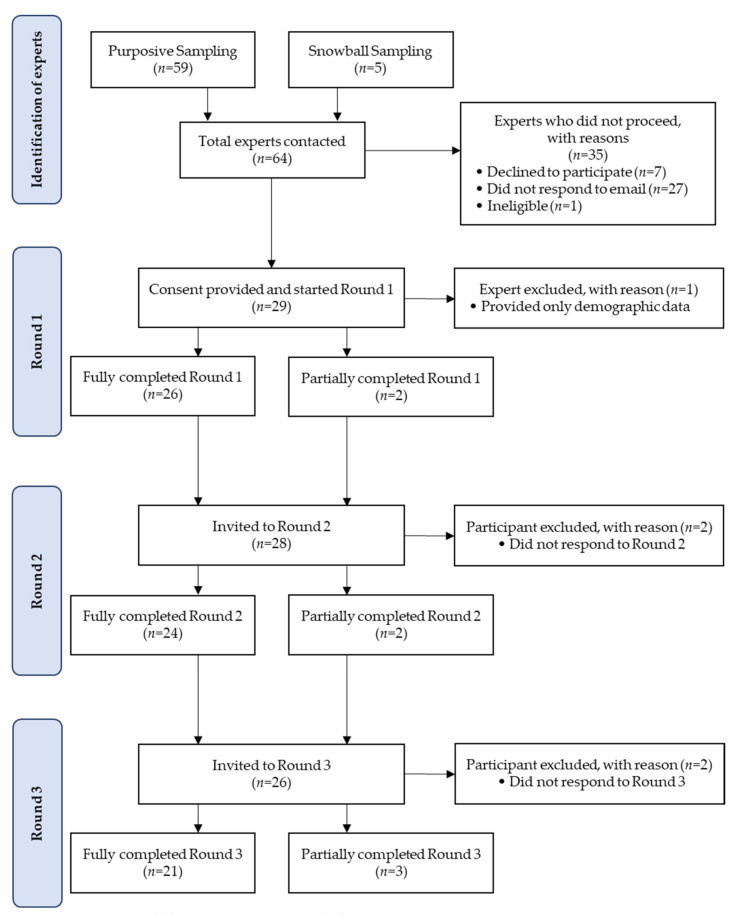
Flow chart of the number of experts from recruitment through to Round 3 of the Delphi study.

**Figure 3 nutrients-13-04087-f003:**
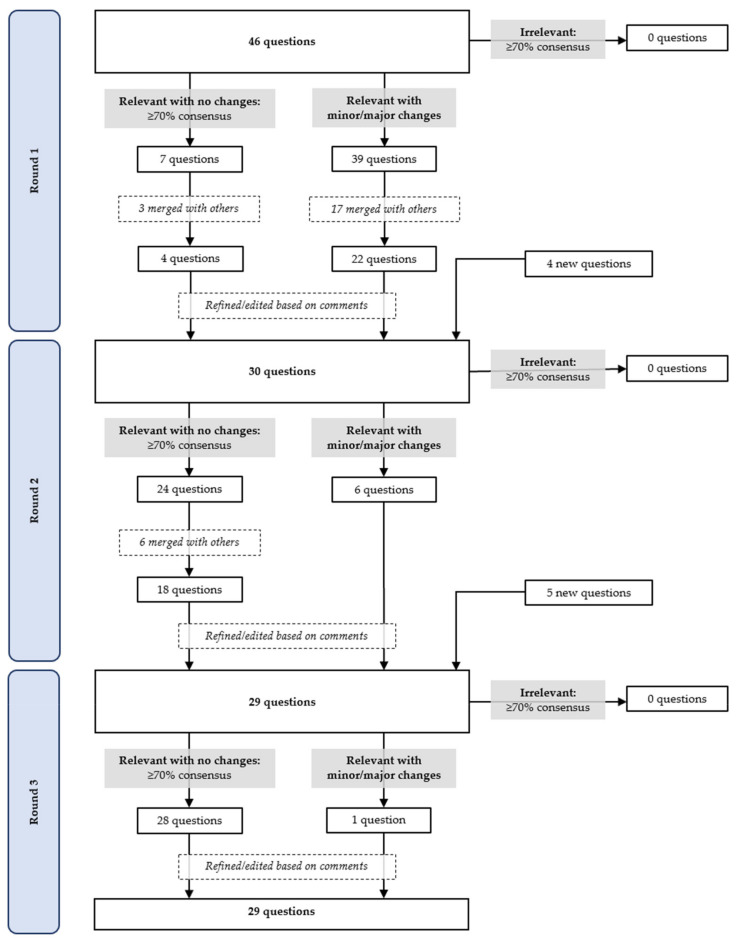
Flow chart of the elimination and addition of questions in the Chrononutrition Questionnaire through Rounds 1 to 3 based on consensus of ratings and feedback from the expert panel.

**Figure 4 nutrients-13-04087-f004:**
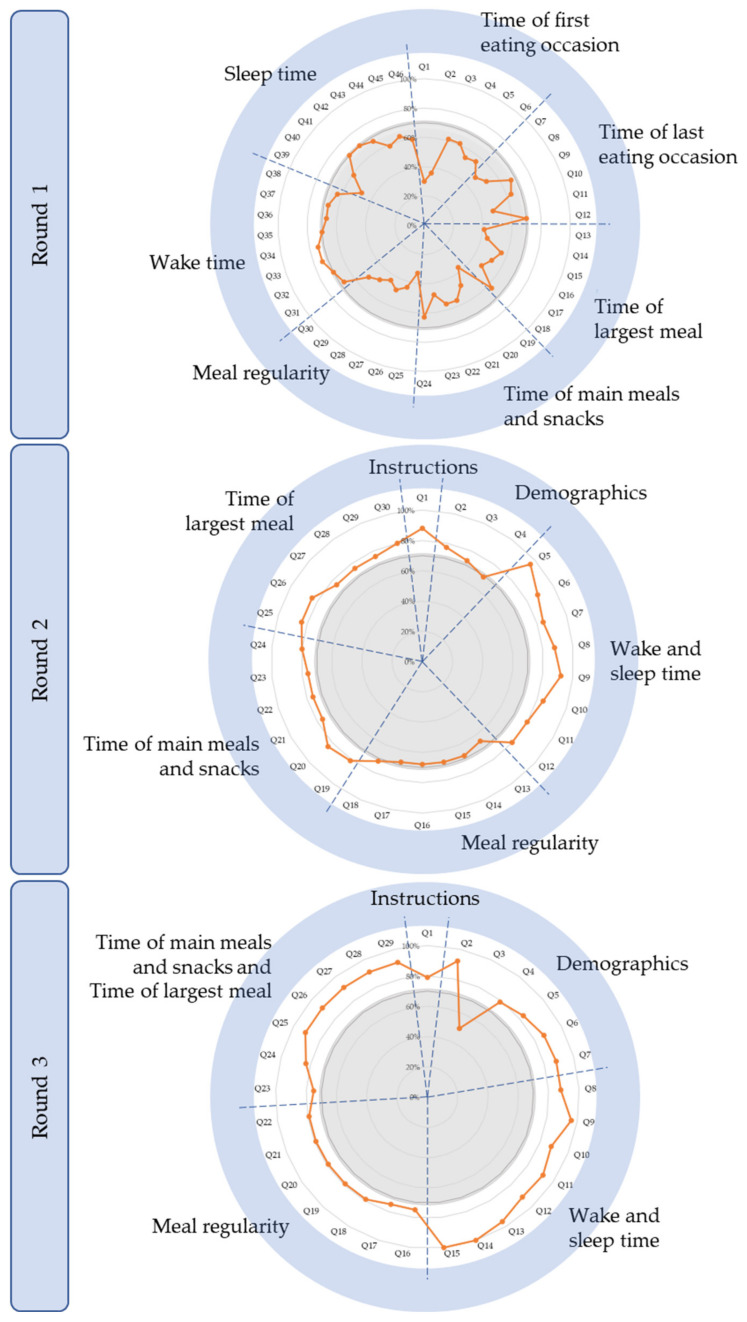
Chrononutrition Questionnaire sections, respective questions, and percentage ratings of relevance by experts in Delphi Rounds 1–3. The dashed blue lines segregate sections within the questionnaire, showing questions within each section; the grey areas represent ≤70% consensus; and the orange lines show ratings of relevance by experts. Note that the order of sections from Rounds 1 to 3 was modified based on feedback from experts to provide the most logical sequence.

**Table 1 nutrients-13-04087-t001:** Quality indicators in reporting Delphi methodology as recommended by Diamond, Grant, and Feldman.

**Study Objective**	Does the Delphi study aim to address consensus?	Yes, by presenting results reflecting the level of consensus amongst members of the expert panel.
**Participants**	How will participants be selected or excluded?	Inclusion criteria: - Researchers with a focus on, and published in the area of circadian biology or chrononutrition. - Able to read and write English. Exclusion criteria: - Unable to commit to the Delphi study period.
**Consensus definition**	How will consensus be defined?	Consensus is defined as ≥70% agreement.
**Delphi process**	Were items dropped? What criteria will be used to determine which items to drop?	No items were dropped, they were merged. Items will be dropped if there is ≥70% consensus on the rating “1: Irrelevant”.
	What criteria will be used to determine to stop the Delphi process or will it be run for a specific number of rounds only?	The Delphi process will run for only three rounds.

**Table 2 nutrients-13-04087-t002:** Demographics of the expert panel.

		*n*	%
**Gender**	Male	5	18
	Female	23	82
**Age**	20–29 years	4	14
	30–39 years	10	36
	40–49 years	8	29
	50–59 years	4	14
	60–69 years	1	4
	≥70 years	1	4
**Current role**	Academic	28	100
	Clinician	0	0
**Highest education level**	Bachelor’s degree	1	4
	Master’s degree	1	4
	Doctorate degree	26	93
**Years of experience in the field of expertise**	1–5 years	7	25
	6–10 years	8	29
	11–15 years	3	11
	16–20 years	3	11
	>20 years	7	25
**Country of work**	Australia	11	39
	Brazil	2	7
	Canada	1	4
	Czech Republic	1	4
	Israel	1	4
	Netherlands	2	7
	United Kingdom	4	14
	United States	6	21

**Table 3 nutrients-13-04087-t003:** Summary of comments provided by the expert panel and response by the research team in Round 1.

Expert Suggestions and Comments		Changes Made or Clarifications
Questionnaire instructions and requirements	Include instructions to participants, with a clear recall period.	New section: “Instructions to participants” as suggested.
	Improve questionnaire format and layout.	As suggested, particularly: - Outcomes “Times of first/last eating occasion” removed, as they are captured by “Times of main meals and snacks”. - Outcomes “Wake time” and “Sleep time” are captured as one outcome, and re-arranged to be before outcomes on “Meal regularity”, “Times of main meals and snacks…”, and “Time of largest meal”.
	Improve choice of wording	As suggested.
Demographic data	Include questions about: - Shift schedule - Start and end times of each shift	New section: “Demographics”, to gather data as suggested.
Outcomes of interest	Instead of “weekdays” and weekends”, use “work/school” and “work-free/school-free days”.	As suggested.
	Consider limitations of asking about sleep/wake patterns only on specific shift and free-day scenarios that not all shift workers have as part of their shift schedules.	Shift and free-day scenarios were based on the MCTQ^Shift^. It is acknowledged that shift workers whose shifts don’t align with these scenarios cannot be chronotyped.
	Determine alarm clock use for waking, as in the MCTQ (waking up without an alarm clock better indicates circadian phase and estimation of chronotype).	Participants asked to state wake up time if able to choose freely (without using an alarm clock and unaffected by children/pets, hobbies) following the ultra-short MCTQ and MCTQ.
	What is the time window for “day of a morning/afternoon/night shift” within which temporal patterns of eating are captured?	Updated definitions.
	Consider that timing of eating occasions “on a work-free day” for shift workers may be affected by the prior day’s shift type.	Updated to “on a work-free day after a work-free day” to minimise influence of the prior day’s shift type on timing of eating occasions.
	Will variation in timing of food intake within the same day type be captured?	Slight variations captured by asking about “typical” times. Otherwise, identified by question on regularity.
	Better capture concept of regularity - Refine definition of “regular”. - Consider Likert scale instead of Yes/No. - Should it be determined separately between each day type, or continuously across all days of the week? - Should regularity of each meal type be determined separately?	- Updated definition - Captured by Likert scale of Never, 25%/50%/75% of the time, Always. - Determined separately between each day type (refer to Discussion). - Determined across all main meals, not separately by meal type (refer to Discussion).
	Consider if one has two meals that are equally large.	Updated to ask about time of largest meal(s).
	Is defining largest meal by portion size too subjective?	No change (refer to Discussion).
	Instead of time of largest meal, consider time when most calories are consumed (drinks and snacks may contain more than a meal).	
	Refine definitions of terms.	As suggested.
Additional outcomes to include	- Sleep latency and quality - Duration of each eating occasion - Temporal variation of sensations such as hunger, appetite, and satiety - Diet composition	Not included.

**Table 4 nutrients-13-04087-t004:** Summary of comments provided by the expert panel and response by the research team in Round 2.

Expert Suggestions and Comments		Changes Made or Clarifications
Questionnaire instructions and requirements	Improve choice of wording.	As suggested.
	Improve questionnaire format and layout.	As suggested, particularly: - Pathway questions to guide participants to questions relevant to them. - Combined outcome of “Time of largest meal” with “Time of all eating occasions”.
Demographic data	Improve definition of “general population” as shift workers are technically within general population.	Removed, as redundant after addition of pathway questions.
	Allow participants to state if they go to both work and school, and the start and end times of each.	As suggested.
	What does the term “school” refer to?	Adults who are studying.
	Include option for non-standard shifts beyond morning/evening/night shifts.	Added option for split shift workers (refer to Discussion).
Outcomes of interest	Better capture concept of regularity: - Consider also capturing regularity of snacks. - Re-consider determining regularity across all days of the week and using Yes/No instead of Likert scale. - Consider using Likert scale of “Always, Usually, Sometimes, or Never” instead of % of time.	- Sub-question about regularity of snacks included. - No change in response to the other suggestions (refer to Discussion).
	What if shift workers have more than one sleep episode in between shifts?	They will be asked to choose times of main sleep, not naps. If they have ≥2 sleeps that are of equal duration, they may choose one, to be validated against data from sleep diaries/actigraphy in a later study.
	One may not be able to freely choose wake up time unaffected by other factors (e.g., children/pets, hobbies).	Participants asked to specify wake up time without alarm clock use only.
	Preference of time window for “day of a morning/afternoon/night shift” within which temporal patterns of eating are captured to be limited by sleep/wake time before and after the shift instead of 12 a.m.–12 a.m. limits for morning shifts and 12 p.m.–12 p.m. limits for afternoon and night shifts.	As suggested.
	Aid identification of eating occasion (≥210 kJ) with a calorie counter.	
	Refine definitions of terms.	
Additional outcomes to include	Food composition, as carbohydrate and fat-rich foods may be relevant in terms of timing of food intake.	Not included.

**Table 5 nutrients-13-04087-t005:** Summary of comments provided by the expert panel and response by the research team in Round 3.

Expert Suggestions and Comments		Changes Made or Clarifications
Questionnaire instructions and requirements	Improve choice of wording.	As suggested.
Demographic data	Provide definition for “work” to include both paid and unpaid work.	As suggested.
	Provide definition for “school”.	
	“General population”: provide an option of “Other” for individuals do not go to work/school and are free to structure their day.	As suggested.
	“Shift work population”: provide more shift options to categorise participants.	
Outcomes of interest	Better capture concept of regularity: - Re-consider using Yes/No instead of Likert scale. - Consider capturing extent of variability in meal timings (e.g., <30 min, 30–60 min, 60–90 min, 90–120 min) between days, or weekdays and weekends.	No change (refer to Discussion).
	Consider if an eating occasion lasts a long duration (e.g., a drink sipped over 3 h).	
	Refine definitions of terms within the questionnaire.	As suggested.
Additional outcomes to include	Are meal breaks at work scheduled or dependent on workload?	Not included (refer to Discussion).
Other	Obtain mixed population feedback about language and burden of the questionnaire.	Considered.

## Data Availability

The data presented in this study are available on request from the corresponding author.

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
