# Peer review of "Content Validation of a Chrononutrition Questionnaire for the General and Shift Work Populations: A Delphi Study"

_nutrients, 2021, doi:10.3390/nu13114087_

Round 1

Reviewer 1 Report

Interesting study, well conducted and well described. The results of the study represent further improvement in the knowledge on this topic.

Minor points:

(i) incorporating a couple of references, in the paragraphs dealing with circadian rhythms (Allada) and chronotype and health (Fabbian)

Circadian Mechanisms in Medicine. Allada R, Bass J. N Engl J Med. 2021 Feb 11;384(6):550-561. doi: 10.1056/NEJMra1802337.

Chronotype, gender and general health. Fabbian F, Zucchi B, De Giorgi A, Tiseo R, Boari B, Salmi R, Cappadona R, Gianesini G, Bassi E, Signani F, Raparelli V, Basili S, Manfredini R. Chronobiol Int. 2016;33(7):863-82. doi: 10.1080/07420528.2016.1176927.

(ii) please complete with all authors references n. 6, 12, 15, 36.

Author Response

Dear Reviewer,

Thank you for your time spent in reviewing our manuscript. Please find our response to your comments in red.

Interesting study, well conducted and well described. The results of the study represent further improvement in the knowledge on this topic.

Point 1: Minor points:

(i) incorporating a couple of references, in the paragraphs dealing with circadian rhythms (Allada) and chronotype and health (Fabbian)

Circadian Mechanisms in Medicine. Allada R, Bass J. N Engl J Med. 2021 Feb 11;384(6):550-561. doi: 10.1056/NEJMra1802337.

Chronotype, gender and general health. Fabbian F, Zucchi B, De Giorgi A, Tiseo R, Boari B, Salmi R, Cappadona R, Gianesini G, Bassi E, Signani F, Raparelli V, Basili S, Manfredini R. Chronobiol Int. 2016;33(7):863-82. doi: 10.1080/07420528.2016.1176927.

Response 1: The first reference has been included in line 39 and 42, and the second reference in line 50.

Point 2: (ii) please complete with all authors references n. 6, 12, 15, 36.

Response 2: To align with MDPI referencing style guideline, we have included the first 10 authors followed by et al and hence the following references have been updated n. 7, 10, 14, 17, 38.

Reviewer 2 Report

Understanding the impact of eating at unusual times and the health issues associated with this is important, and the authors have conducted a good piece of research to create a useful questionnaire. I only have some queries:

Fig 1 - what is the different between outcome 3 largest meal and outcome 4 main meal? If shift workers eat almost continuously then they may not have what we would term as a main/large meal as food would be fairly evenly distributed throughout the shift.

Line 174 - why choose 70% as the cut-off for acceptance?

Shame no clinicians were included - Table 2 as they often work shifts and their health suffers

Author Response

Dear Reviewer,

Thank you for your time spent in reviewing our manuscript. Please find our response to your comments in red.

Understanding the impact of eating at unusual times and the health issues associated with this is important, and the authors have conducted a good piece of research to create a useful questionnaire. I only have some queries:

Point 1: Fig 1 - what is the different between outcome 3 largest meal and outcome 4 main meal? If shift workers eat almost continuously then they may not have what we would term as a main/large meal as food would be fairly evenly distributed throughout the shift.

Response 1: Thank you for allowing us to clarify this. In the questionnaire, we have defined “main meal” as breakfast, lunch, dinner, and/or supper, and “largest meal” based on portion size. These details have been added in the figure legend for Figure 1 (line 108).

Identifying “main meals” will allow us to determine the temporal spread of food intake across the day, with participant-identified main meals suggesting at times at which caloric load is greater, while “largest meal” seeks to further determine the time of the meal where caloric load is the greatest in the day (i.e., the largest within the main meals identified). Please note this has been referred to in the manuscript on lines 405-409. We had chosen portion size to be an arbitrary indicator of energy load as a way to overcome the layperson’s lack of knowledge regarding the calorie content of food, since it has been shown that larger portion sizes results in greater energy consumption[1].

Ultimately, we plan to test the validity of our assumptions of the terms “main meals” and “largest meal” as indicators of relative caloric load in a subsequent study by comparison against food diaries (that contain calorie information of food consumed) completed within the same time frame.

Lastly, we understand that shift workers may have fairly evenly distributed food intake (i.e., main meals indistinguishable from snacks); whether this is the case will be tested in our subsequent study of questionnaire results against food diaries. At the same time, we have chosen to still include these terms as our questionnaire will also be completed by non-shift workers. Please note these points have been referred to in the manuscript on lines 441-445.

Point 2: Line 174 - why choose 70% as the cut-off for acceptance?

Response 2: Despite one of the objectives of the Delphi study being to achieve consensus, there is no universal agreement regarding an ideal level, with a range of 51-80% in previous studies, also dependent on the importance of the research topic[2,3]. As such, we chose a 70% cut-off, the same as previously used by other studies on questionnaire development in the health arena[4,5]. Please note this has been stated in the manuscript on lines 176-178.

Point 3: Shame no clinicians were included - Table 2 as they often work shifts and their health suffers

Response 3: We have planned to test this questionnaire for reliability and construct validity amongst individuals within the general population and shift workers, of which clinicians who work shift duty will be recruited. An addition referring to this has been included in line 459-460.

Reviewer 3 Report

The manuscript does a good job of addressing what it is intended to accomplish.

It is a bit disappointing that nothing beyond fairly basic analysis was undertaken. The authors hopefully will follow up in subsequent work to establish stringent criteria for reliability, validity, and factor structure. Ultimately, application of confirmatory factor analysis would be ideal.

Author Response

Dear Reviewer,

Thank you for your time spent in reviewing our manuscript. Please find our response to your comments in red.

The manuscript does a good job of addressing what it is intended to accomplish.

Point 1: It is a bit disappointing that nothing beyond fairly basic analysis was undertaken. The authors hopefully will follow up in subsequent work to establish stringent criteria for reliability, validity, and factor structure. Ultimately, application of confirmatory factor analysis would be ideal.

Response 1: Thank you for your comment, we have planned for this questionnaire to undergo testing of test-retest reliability as well as construct validity against food diaries, sleep diaries, and wrist accelerometers next year. We will take your suggestions into consideration in planning of subsequent work.